# Purchase Channels and Motivation for Exercise in the Slovenian Population: Customer Behavior as a Guarantee of Fitness Center Sustainability

**DOI:** 10.3390/bs13060447

**Published:** 2023-05-28

**Authors:** Vojko Vuckovic, Ivan Cuk, Sasa Duric

**Affiliations:** 1Faculty of Sport, University of Ljubljana, 1000 Ljubljana, Slovenia; vojko.vuckovic@fsp.uni-lj.si; 2Faculty of Sport and Physical Education, University of Belgrade, 11000 Belgrade, Serbia; 3Liberal Arts Department, American University of the Middle East, Al-Egaila 54200, Kuwait; sasa.duric@aum.edu.kw

**Keywords:** EMI-2 questionnaire, fitness center management, young Slovenians, motivation for physical exercise

## Abstract

The sustainability of fitness centers depends on two factors: member recruitment and retention, which is why these factors have received attention in recent decades. Temporal trends in fitness center membership purchase channels from 2016 to 2022 and motivation for exercise in 2022 in the Slovenian general population were investigated. The sample included 3419 participants, including 3131 participants (age 31.03 ± 11.31 years, 1430 females) and 288 participants (age 29.39 ± 10.43 years, 110 females) for the first and second objectives, respectively. Data were assessed using a web-based recruitment questionnaire and the EMI-2 motivation questionnaire. Traditional advertising strategies such as radio and flyers are the least effective (only 0.9% of memberships in 2022), while more sophisticated advertising strategies such as the internet and social media are becoming increasingly important in the advertising world (26.6% of memberships in 2022). On the other hand, word of mouth is the most influential method, attracting 51.3% of new members. Females, older members, and Eastern Slovenians were more motivated to exercise by health and esthetic motives, and males and younger members by challenge and competition. Fitness center management should focus on providing the best possible quality of service, tailoring it to the age, gender, and motivation of customers.

## 1. Introduction

Nowadays, people invest a lot of time and resources in a healthy lifestyle, which is reflected in the growth of the fitness industry. With an estimated annual revenue of $87 billion in 2018, it is one of the world’s fastest-expanding industries [1]. According to public reports, 11% of citizens in the European Union reported being a member of a fitness center [2], while in the United States, 20.6% of the population are members of a health/fitness club/gym [3]. In Norway, for example, the number of fitness centers has almost tripled in the last decade, from 477 to 1228 [4]. Fitness center membership is not only associated with greater health responsibility and health-related behaviors [5], but also with some psychological and social benefits, such as reduced loneliness and social isolation [6] and higher levels of satisfaction [7]. Therefore, fitness centers have become an important venue for physical activity and exercise [8]. However, the sustainability of fitness centers relies on membership recruitment (attracting new customers) and retention (maintaining membership for as long as possible). Both are important, which is why these factors have received special attention in recent decades [9].

To attract new customers, it is important to know what factors influence customers’ purchase intention and what motivates potential customers to exercise. Previous studies have investigated the factors influencing customers’ intention to purchase fitness center membership [10,11,12,13]. These studies have shown that many factors can influence purchase intention, such as product availability or the number of branches, price and activity classes offered [12], brand image [13], facility [10], service quality [11], etc. Previous studies have shown that both traditional and social media advertising are positively related to purchase intention, with social media advertising having a greater impact [14]. A study by Alalwan et al. [15] showed that factors such as hedonic motivation, performance expectancy, informativeness, and perceived relevance were all positively related to purchase intention for products advertised on social media. Fitness and health-related advertising accounted for 9% of total advertising spend in the United States in 2017 [16]. On the other hand, there is an opinion that increasing retention levels by building and maintaining relationships with existing customers will lead to future purchases [17]. This is also supported by a study by Schmittlein [18], which states that the existing customer base is an important strategic asset, as the cost of acquiring new customers is five times higher than that of retaining existing customers. Therefore, relationship marketing is very important to identify specific customer segments and establish, maintain, and improve the relationship between customers and companies, thus emphasizing customer retention greatly [9]. Although marketing is of tremendous importance in the fitness industry, there is little published research on the subject. For example, no research has been found on the effect of various marketing interventions. Clubs would benefit greatly from having more insight into the effect of the various marketing tools available. For example, analysis of customer databases could provide a lot of information [19]. Therefore, further research in this area is needed.

As a second pillar of fitness center sustainability, member retention can be observed through member motivation, a very important psychological factor for participation in regular exercise. The goals of exercise and physical activity range from improving physical fitness and health to enhancing talents and skills [20]. A study conducted in the United States found that the main motivators for gym membership were appearance, functioning, and health [21]. Our previous study conducted with college students in Slovenia showed that female students considered weight management as the main motivating factor. In contrast, male students indicated fun, challenge, social recognition, belonging, competition, and strength and endurance as the main motivators for exercise [22]. Although there is a huge population of fitness center members worldwide, research on their exercise behavior is limited in quantity and quality [23]. Therefore, it is important to obtain comprehensive knowledge about the health-related behavior of regular fitness center members [24]. Consequently, there is a need to explore the motivation for exercise in the general population, especially in Central Europe, due to a lack of research data.

For these reasons, we designed a study with two objectives: (1) to investigate the temporal trends of fitness center membership purchase channels from 2016 to 2022 and (2) to investigate the motivation for exercise in 2022 in the Slovenian general population. We cannot hypothesize on the first objective of the study due to inconclusive results in the literature and a variety of variables that have been examined. However, regarding motivation for exercise, we hypothesize that social recognition and competition are more important for male and younger participants, while weight management and appearance are more important for female and older participants.

This is the first study of its kind in Slovenia, where people generally maintain a healthy lifestyle and children are among the most active in the world [25,26]. The fact that Slovenia is one of the few countries that have been conducting physical fitness testing of the entire population for over 40 years [27,28] speaks to people’s high awareness of the importance of physical activity for health. In this respect, conducting such research among the population in this part of Europe is particularly important.

## 2. Materials and Methods

### 2.1. Participants

The total sample included 3419 participants, members of the 4P Fitness Center in Ljubljana (Western Slovenia) and Novo Mesto (Eastern Slovenia). The sample consisted of two subsamples: the first subsample of 3131 participants (age 31.03 ± 11.31 years, 1430 females and 1701 males) was involved in assessing reasons and advertising channels for new membership each year from 2016 to 2022 (on average 447 participants per year), while the second subsample of 288 participants (age 29.39 ± 10.43 years) reported on the motivation for exercise in 2022. Participants were later classified by gender (110 females and 178 males), by age: youth ≤ 24 years (n = 134) and adults > 24 years (n = 154), and by geographic location: Eastern Slovenia (n = 53) and Western Slovenia (n = 235).

### 2.2. Instruments

Temporal trends in membership purchase channels from 2016 to 2022 were assessed using a web-based questionnaire that included five questions about advertising channels and reasons for joining the fitness center (see Figure 1). The Exercise Motivations Inventory-2 (EMI-2), developed by Markland and Ingledew [29] assessed exercise motivation among fitness center members. The questionnaire consisted of 51 items, each scored on a 6-point Likert scale ranging from 0 to 5. The higher the score, the higher the motivation to exercise. The above scales formed 14 scales: stress management, revitalization, enjoyment, challenge, social recognition, affiliation, competition, health pressures, ill-health avoidance, positive health, weight management, appearance, strength and endurance, and nimbleness. The scales were determined according to the EMI-2 scale scoring key [30]. The EMI-2 questionnaire is a valid instrument for assessing a wide range of motives for participation in athletic activities in adults and is suitable for both athletes and non-athletes [29]. The reliability of the 51 EMI-2 items was also confirmed in this study by measuring Cronbach’s Alpha coefficient (see Results). 

### 2.3. Procedure

Reasons and advertising channels for joining were assessed in 2016, 2017, 2018, 2019, 2020, 2021, and 2022. Motivation to participate in exercise was assessed in 2022. Participants first signed an informed consent form and then completed questionnaires through an online platform called (www.1ka.si). This study was conducted in accordance with the Declaration of Helsinki. All participants provided written informed consent before participating in the study, and the Ethics Committee of the University of Ljubljana granted ethical approval for data collection (No. 2021-19).

### 2.4. Statistical Analysis

The reliability of the EMI-2 scales was tested using Cronbach’s Alpha test. The means of the 14 scales were used as dependent variables, and participant characteristics (female vs. male, youth vs. adults, Eastern Slovenia vs. Western Slovenia participants, marital status, and type of training) as independent variables. Due to the ordinal and nominal nature of the data, the nonparametric Mann–Whitney U test was used in this study. Alpha was adjusted with a Bonferroni correction. All statistical tests were analyzed using the Statistical Package for the Social Science software (SPSS Statistics 23; IBM, Armonk, NY, USA). The alpha level was set at 0.05.

## 3. Results

Figure 1 shows the trends in new member recruitment from 2016 to 2022. The solid line with full squares at the bottom of the figure shows that a very small percentage of members joined because they learned about the fitness center through radio and flyers (0.9% in 2022–1.6% in 2016). This trend was fairly stable as the least effective in attracting new members. »Other« as a reason for joining peaked in 2016 (16.3%) and decreased to only 4.3% in 2022. Another reason for (re)joining the fitness center was that the practitioner had been there before, with 54% in 2016 and 16.5% in 2022. Two important advertising strategies that showed an increasing trend from the beginning almost to the end of the period were the internet, social media, and word of mouth. Internet and social media were a source of information for 8.4% of practitioners in 2016. This number has increased significantly over the years, with a peak in 2020 (31.8%) and a slight decrease thereafter (26.6% in 2022). Word of mouth is another source of information that appears to be increasing from 2016 (19.6%) to 2022 (51.3%). Other reasons and channels for joining the fitness center, such as randomly walking past the fitness center, accounted for 16.3% of new members in 2016 and only 4.7% in 2022.

The association between genders and age groups for the advertising strategies used is shown in Table 1.

All correlation coefficients for all advertising strategies were positive and significant except for radio and flyers, which were negative and not significant. The same nature of the relationship was found for the age group factor. In addition, the correlation coefficient for internet and social media was not significant for the age group.

The scales, sample items, and Cronbach’s Alpha for EMI-2 are shown in Table 2. 

All Cronbach’s Alpha coefficients were high, indicating high reliability of the items used (0.711 ≤ a ≤ 0.908). 

The differences between female and male participants in mean ranking scores on exercise motivation are shown in Figure 2. The results of the Mann–Whitney U test revealed that five motivational scales differed in importance between the genders as motivational factors for exercising. Social recognition and competition were more important for males, while ill-health avoidance, positive health, and weight management were the main motivators for women’s exercise.

Figure 3 shows the differences between the youth and adult participants regarding mean ranking scores on exercise motivation. The analysis identified six motivational scales as differentially important motivating factors for exercise between the different age groups. Challenge and competition were more important for youths, while revitalization, ill-health avoidance, positive health, and appearance were the main reasons for exercising in adults.

Table 3 shows the differences in mean ranking scores between participants of the same gender but different age category or same age category and different gender.

Mean ranking scores on exercise motivation and the differences between participants with different demographics are shown in Figure 4.

Figure 4 shows that six motivational scales differ significantly between practitioners from Eastern and Western Slovenia. These are stress management, revitalization, positive health, weight management, strength endurance, and nimbleness. It is interesting to note that the above motivational scales are significantly higher among participants from Eastern Slovenia.

Additional analyses were performed on participants regarding their marital status and type of training. The results showed that single participants were more motivated to exercise because of their affiliation, while for married/in relationship participants, revitalization, ill-health avoidance, positive health, and appearance were considered more important factors for exercise. Regarding the type of training, there were differences in three scales: ill-health avoidance, positive health, and weight management. Ill-health avoidance and positive health were the most important motivating factors for group training participants, while weight management was the most important for personal training participants.

## 4. Discussion

We designed the present study to investigate membership recruitment and retention by examining temporal trends of membership purchase channels in fitness center from 2016 to 2022 and leisure-time motivation for exercise in 2022 in the Slovenian general population.

As mentioned earlier, the sustainability of fitness centers is based on two factors: membership recruitment and retention. To investigate membership recruitment, we examined the temporal trends of membership purchase channels from 2016 to 2022, as shown in Figure 1. It can be seen that the least effective method of recruiting new members was through radio and flyer advertising (between 0.9% and 1.6% of members). A more detailed analysis reveals that older people are more receptive to this type of advertising, but at a very low percentage (the highest value was 4.3% of members in 2022). It is noticeable that older people still rely on traditional advertising methods, while younger people are turning more to the internet and newer technologies. A study from Ireland yielded similar results, with the lowest percentage of participants relying on leaflets and newsletters to join the fitness center (12% and 9%, respectively) [9]. These results suggest that this is not the best strategy for advertising a fitness center and that owners and managers should not invest time and money in such traditional advertising strategies but instead focus on other newer strategies. This is supported by the study of Fogel and Ustoyev [16], which concludes that marketers can consider novel social media advertising that promotes deposit contract information to attract potential customers to join a fitness club or gym. This is also supported by the fact that the internet and social media showed an increasing trend from the beginning (8.4% of members in 2016) to almost the end (26.6% of members in 2022) of the observation period. However, despite the increasing trend, there was a slight decrease in 2021 (22.6%) compared to 2020 (31.8%), quite in contrast to the “being a member before” reason for joining. This reason generally declined (54% to 16.5% of members in 2016 and 2022, respectively). However, looking at the trend marked with white diamond-shaped symbols, there was a significant increase in 2021 compared to 2020. This was more pronounced among adults (20.2% in 2020 and 40.1% in 2021), while a similar trend was observed among younger people (10.8% in 2020 and 20.8% in 2021). It should be noted that the trends between older and younger people and between genders show a high and significant correlation (see Table 1). We can assume that people were concerned about their health after the pandemic and reminded themselves to go to the gym. This statement can be supported by Kahneman’s [31] theory of cognitive ease and strain, which states that the stress responses triggered by the perception of viruses such as COVID-19 may make individuals more inclined to process information relevant to their situation. Furthermore, previous studies have shown that the presence of COVID-19 has been shown to influence the decision-making process [32,33]. In addition, gyms or fitness centers were considered places with a high risk for COVID-19 virus transmission [34]. On the other hand, previous studies have also shown that exercise can improve the immune system by increasing the number of immune cells and strengthening the body’s defenses against the virus, thus providing a cost-effective, practical, and non-pharmacological way to deal with COVID-19 [35]. This direct link between exercise and health has encouraged consumers to start exercising, regardless of how much they exercised before the pandemic [36].

The most surprising and potentially important finding was that becoming a new member through a friend (or word of mouth) as a reason for joining increased from 19.6% of members in 2016 to 51.3% of members in 2022. Previous studies show that word of mouth is one of the most influential communication channels in the market [9]. Word of mouth is considered more credible than marketer-initiated communications because it is perceived to have gone through the unbiased filter of “people like me”. At a time when trust in institutions is waning, research shows that the influence of word of mouth is growing stronger [37]. This is consistent with the increasing trend throughout the observation period and suggests that quality of service is the most important factor in attracting new customers. Another study by Natus et al. [38], which aimed to identify the main characteristics of perceived value in the construct of “word of mouth” in a fitness center, showed that the dimensions of service (emotional aspects, image/reputation, and price) were related to word of mouth, thus positively influencing customers’ behavior. Image and reputation were highlighted compared to the other dimensions and had a higher coefficient value. Another study showed that rapport was positively associated with word of mouth in the fitness industry [39]. Hence, these dimensions are of particular importance when providing a higher quality of service.

There were also other reasons for joining—16.3% of members in 2016 and 4.7% in 2022. It should be further investigated what these other reasons could be, but given the decreasing trend and the very low percentage in 2022, perhaps not too much importance should be given to this investigation. 

From the obtained results, we can conclude that, in order to attract new members, fitness center owners and management should not invest in “old-fashioned” traditional advertising methods such as radio and flyers, but instead should direct their advertising strategies towards more modern strategies such as the internet and social media, and above all, spend time and energy on providing the best possible service to their customers.

Regarding member retention, we examined customers’ leisure-time motivation for exercise in 2022 using the EMI-2 questionnaire. In our previous study [22], we used the mentioned questionnaire for the first time in Slovenia. Therefore, we performed a factorization of the questionnaire and found that the obtained factors were very similar to the ones proposed by the authors of the questionnaire [29]. Our first study on students showed that the original scale had high reliability [22]. In the present study, our results also show that the original scale has high reliability for the general Slovenian population; all 14 motivational scales have a Cronbach’s Alpha greater than 0.7, while 8 of them are greater than 0.8. Finally, the results of the EMI-2 questionnaire showed that there were significant differences in some motivation scales between females and males, youths and adults, and participants from Eastern and Western Slovenia (Figure 2).

The Mann–Whitney U test showed that five motivational scales were of different importance to females and males. Social recognition and competition were more important for males, which is in line with previous studies [40,41,42,43,44,45] and our previous study conducted with the student population [22]. However, two studies from Brazil [46] and the Philippines [47] showed that competition was among the least important reasons for exercising. On the other hand, our previous study showed that male students are more often motivated by fun than female students, which is not the case in the present study. As for female members, the results of the present study revealed that ill-health avoidance, positive health, and weight management were the main motivators for exercise. These findings are also consistent with several previous studies [22,40,41,42,44,45,48,49,50]. However, it is worth noting that our earlier study with university students did not confirm positive health findings in the female population [22]. Thus, it is clear that positive health and health in general become more important as we age. To support this claim, the results showed that revitalization, ill-health avoidance, positive health, and appearance were the main reasons for exercising in adults. In contrast, challenge and competition were more important for youths (Figure 3). It has already been shown that motives such as challenge and affiliation are significantly higher among participants in competitive activities [44] and that competition is positively associated with higher levels of intrinsic motivation [51]. Therefore, it can be speculated that male and younger members were more driven by intrinsic motivation and may have been involved in some competitive activities. This was also supported by a Norwegian study by Larsen et al. [52], who concluded that motives for exercise participation in gyms depend on individual characteristics and that motives for exercising in gyms differ by gender and change with age. For example, they showed that the importance of enjoyment and competition decreased with age in men, while women seemed to place more importance on vitality as they aged. These results are exactly in line with what we confirmed with the separate analysis in Table 3, which shows the differences in mean rankings between participants of the same gender but different age category or the same age category and different age group. In addition, it appears that health, weight management, appearance, and nimbleness increase with age in women, whereas enjoyment, challenge, social recognition, affiliation, and competition decrease with age in men.

Similar to the different age groups, there were significant differences between members from Eastern and Western Slovenia for six motivational scales (Figure 4). Stress management, revitalization, positive health, weight management, strength endurance, and nimbleness were significantly higher among participants from Eastern Slovenia. We can only speculate that people from this region care more about these factors because they live in a smaller community, consequently leading a more “peaceful” lives and may have more time to take care of their health. A comparison can be made in part with a study by Promsaan and Rattanapunya [53], who used the EMI-2 questionnaire and showed that older people in rural areas had significantly higher motivation to exercise in all aspects than those in urban areas. The authors concluded that health promoters should encourage exercise promotion among older people in urban areas. Perhaps this can also be applied to our results, and we can conclude that exercise promotion should have a higher priority in Ljubljana, the capital of Slovenia. Nevertheless, we should limit our conclusions regarding geographical criteria due to the relatively disproportionate sample size.

### Strengths and Limitations of the Study

This is the first study to investigate both factors important to fitness center sustainability: membership recruitment in terms of purchase channels and leisure-time motivation for exercise in the general Slovenian population. An additional value is the insight into the data from the temporal trends of fitness center membership recruitment. This provided us with data to compare and predict future outcomes, providing us with deeper insight into the direction of future advertising and working strategies. 

One of the limitations of this study is that motivation to exercise was assessed only in 2022. A temporal trend in motivation would be desirable to provide a comprehensive overview of motivational factors throughout the observation period. This would also help fitness center management to predict the future motivational factors of their future customers so that they could benefit from such an analysis and provide the best possible service in the future. Furthermore, future studies should be conducted in a wider geographical area. In addition, the different sample size in Eastern and Western Slovenia could potentially bias the results, somewhat limiting their interpretation. Finally, due to the nominal and ordinal nature of the data, an analysis that would reflect the interaction effects of multiple independent variables on the individual dependent variables and allow for more in-depth statistical analysis was not possible.

## 5. Conclusions

We believe that the results of the present study are of great theoretical and, above all, practical importance. They provide insight into the temporal trends of membership purchase channels and member motivation, which may help fitness center owners and management design their promotional and operational strategies to maintain the sustainability of their fitness centers in terms of membership recruitment and retention.

It has been shown that traditional advertising strategies such as radio and flyers are the least effective, but on the other hand, more sophisticated advertising strategies such as the internet and social media are becoming increasingly important in the advertising world. However, the results of the present study show that word of mouth is the most influential method for attracting new members, which indicates that fitness center management should focus their energy on providing the best possible quality of service. 

In terms of motivation to exercise, our study showed that females, older members, and members from Eastern Slovenia were more likely to be driven to exercise by health and esthetic motives. In comparison, males and younger members were motivated by challenge and competition. This is an indication that targeting customers of different gender, age, and demographic characteristics should be tailored to their needs in order to maintain or exceed past operational and financial performance.

### Managerial Implications

Considering the implications of our research for fitness centers’ operations, we advise fitness managers to invest more in newer marketing solutions such as digital marketing and social media. In addition, service quality and satisfaction with the fitness center are very important, as the trend in our study clearly shows that word of mouth and e-promotion are becoming the most important ways to attract customers. Our results also show that gym managers should consider that different customers have different motivations to exercise, and if we want them to stay active, we need to communicate with them differently. In this way, they would increase customers’ repurchase intention and generate more revenue. This is extremely important in the fast-growing and very competitive health and fitness industry.

## Figures and Tables

**Figure 1 behavsci-13-00447-f001:**
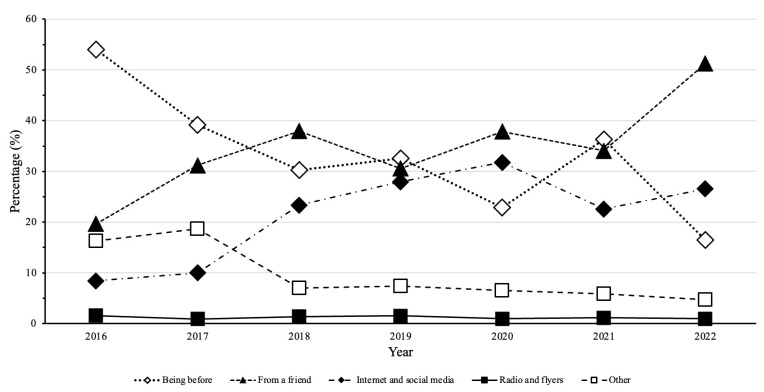
Temporal trends of membership purchase channels in 4P fitness center from 2016 to 2022.

**Figure 2 behavsci-13-00447-f002:**
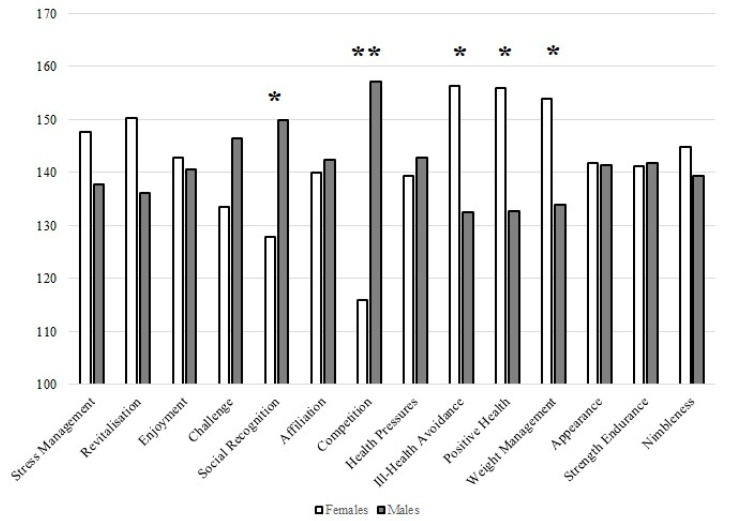
Results of the Mann–Whitney U test for the mean ranking scores on exercise motivation between female and male participants. Legend: * *p* < 0.05, ** *p* < 0.01.

**Figure 3 behavsci-13-00447-f003:**
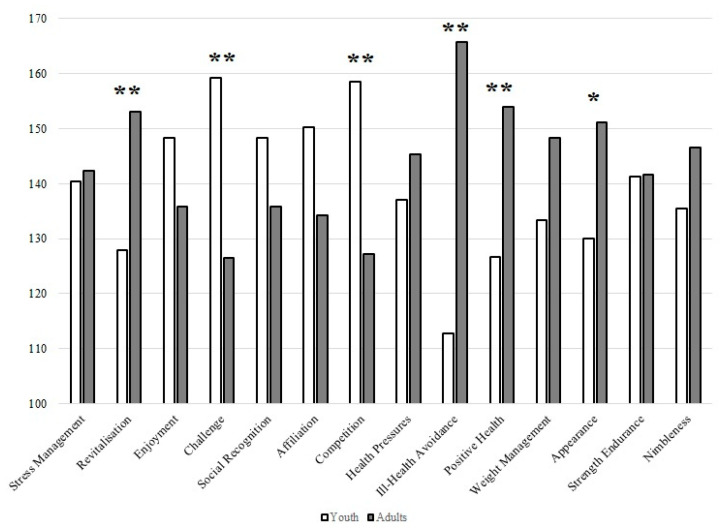
Results of the Mann–Whitney U test for the mean ranking scores on exercise motivation between youths and adults. Legend: * *p* < 0.05, ** *p* < 0.01.

**Figure 4 behavsci-13-00447-f004:**
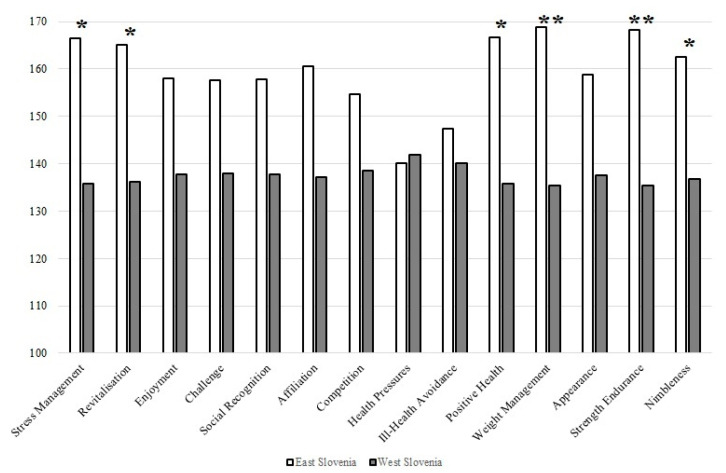
Results of the Mann–Whitney U test for the mean ranking scores on exercise motivation between participants from Eastern and Western Slovenia. Legend * *p* < 0.05, ** *p* < 0.01.

**Table 1 behavsci-13-00447-t001:** Pearson’s correlation analysis between genders and age groups for each advertising strategy.

Factor	I Was Before	Through a Friend	Internet and Social Media	Radio and Flyers	Other
Gender	0.986 **	0.884 **	0.901 **	−0.549	0.775 *
Age group	0.914 **	0.804 *	0.696	−0.401	0.829 *

* *p* < 0.05, ** *p* < 0.01.

**Table 2 behavsci-13-00447-t002:** Scales, sample items, and Cronbach’s Alpha coefficients of the Exercise Motivation Inventory-2 (EMI-2).

Scale	Sample Item	No. of Items	α
Stress Management	To give me space to think	4	0.842
Revitalisation	To recharge my batteries	3	0.711
Enjoyment	Because I feel at my best when exercising	4	0.843
Challenge	To give me goals to work towards	4	0.760
Social Recognition	To show my worth to others	4	0.844
Affiliation	To spend time with friends	4	0.874
Competition	Because I like trying to win in physical activities	4	0.908
Health Pressures	Because my doctor advised me to exercise	3	0.718
Ill-Health Avoidance	To prevent health problems	3	0.789
Positive Health	Because I want to maintain good health	3	0.851
Weight Management	To stay slim	4	0.777
Appearance	To have a good body	4	0.739
Strength and Endurance	To develop my muscles	4	0.846
Nimbleness	To maintain flexibility	3	0.820

**Table 3 behavsci-13-00447-t003:** Results of the Mann–Whitney U test for the mean ranking scores on exercise motivation between youths and adults of the same gender (left panel) and between females and males of the same age category (right panel).

Scales		Females		Males			Youths		Adults	
	Age Category	Mean Rank	*p*-Value	Mean Rank	*p*-Value	Gender	Mean Rank	*p*-Value	Mean Rank	*p*-Value
**Stress Management**	Youths	51.80	0.507	89.04	0.799	Female	66.25	0.769	81.51	0.316
	Adults	55.79		87.10		Male	64.26		74.17	
**Revitalization**	Youths	44.36	**0.003**	84.37	0.371	Female	63.38	0.700	86.86	**0.024**
	Adults	61.84		91.13		Male	65.96		70.81	
**Enjoyment**	Youths	51.29	0.412	97.26	**0.024**	Female	59.02	0.158	83.59	0.142
	Adults	56.20		80.02		Male	68.54		72.86	
**Challenge**	Youths	55.70	0.609	104.02	**0.001**	Female	54.82	**0.017**	78.67	0.711
	Adults	52.62		74.19		Male	71.03		75.95	
**Social Recognition**	Youths	51.03	0.371	96.70	**0.035**	Female	52.63	**0.004**	75.98	0.822
	Adults	56.42		80.51		Male	72.33		77.64	
**Affiliation**	Youths	47.38	**0.046**	103.14	**0.001**	Female	51.73	**0.002**	87.68	**0.018**
	Adults	59.39		74.96		Male	72.86		70.30	
**Competition**	Youths	50.39	0.276	108.37	**0.001**	Female	42.86	**0.001**	74.47	0.576
	Adults	56.94		70.45		Male	78.12		78.59	
**Health Pressures**	Youths	49.66	0.189	87.83	0.966	Female	60.88	0.332	78.76	0.695
	Adults	57.53		88.15		Male	67.44		75.89	
**Ill Health Avoidance**	Youths	41.35	**0.001**	72.51	**0.001**	Female	68.45	0.418	87.08	**0.023**
	Adults	64.29		101.35		Male	62.96		70.67	
**Positive Health**	Youths	43.45	**0.001**	84.07	0.330	Female	64.93	0.986	91.07	**0.001**
	Adults	62.58		91.38		Male	65.04		68.17	
**Weight Management**	Youths	46.71	**0.027**	86.98	0.804	Female	65.50	0.907	88.51	**0.011**
	Adults	59.93		88.88		Male	64.70		69.78	
**Appearance**	Youths	42.68	**0.001**	87.96	0.993	Female	55.55	**0.026**	84.99	0.074
	Adults	63.21		88.03		Male	70.60		71.98	
**Strength Endurance**	Youths	49.58	0.174	91.94	0.328	Female	59.68	0.203	81.64	0.294
	Adults	57.59		84.60		Male	68.15		74.09	
**Nimbleness**	Youths	43.45	**0.001**	92.38	0.282	Female	56.33	**0.040**	88.14	**0.012**
	Adults	62.58		84.22		Male	70.14		70.01	

## Data Availability

The data presented in this study are available on request from the corresponding author.

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
