# Peer review of "Purchase Channels and Motivation for Exercise in the Slovenian Population: Customer Behavior as a Guarantee of Fitness Center Sustainability"

_behavsci, 2023, doi:10.3390/bs13060447_

Round 1
Reviewer 1 Report
Please see the attached file

English is fine and minor editing of the English language is required. However, I am not a mother language reviewr. Thus, I suggest you request a mother language consultancy to ensure an excellent English revision.
Author Response
Dear Authors,
I am pleased for the opportunity to review your article. Here there are some suggestions that I would like to share with you:
Answer: Thank you. Please note that all changes have been made to the main document according to your comments using the "Track Changes" feature.
INTRODUCTION:
Well structured and well written.
Answer: Thank you very much.
METHODS:
In the methods section, you declared that 53 subjects were from Eastern Slovenia and 235 from Western Slovenia. Why did you find this imbalance between the Eastern and Western parts? Please explain this and what you did to manage this data (Can this imbalance influence your outcome? If yes, what did you do to avoid bias?)
Answer: Thank you for this observation. It is true that the sample was unequal. We even mentioned this in the last sentence of the discussion: “Nevertheless, we should limit our conclusions regarding geographical criteria due to the relatively disproportionate sample size” (lines 646-647). It was a sample that we had access to. The nonparametric Mann-Whitney U test was used as for other variables. Please note that this was not the primary goal of our study, yet we added the following sentence to the study limitations: “In addition, the different sample size in Eastern and Western Slovenia could potentially bias the results, somewhat limiting their interpretation” (lines 661-662).
STATISTICAL ANALYSIS:
You declare that the nature of your data is non-normal, but have you verified that? If yes, please explain the statistical procedures. If not, please motivate it.
Answer: Thank you very much for the comment. Actually, we wanted to justify the use of the nonparametric Mann-Whitney U test, which we did in the sentence before. The normality of the data was not the issue here at all. After reviewing your comment, we have determined that the sentence you mentioned is absolutely unnecessary and have therefore removed it. To provide more clarity, we have changed the previous sentence to read, "Due to the ordinal and nominal nature of the data, the nonparametric Mann-Whitney U test was used in this study" (lines 275-276).
RESULTS:
Please explain what the word “OTHER” means. Please provide any example of what you have inserted in the voice “OTHER” (You can put it in the sentence using an expression like “such as” or similar, and provide some example.
Answer: Thank you for the comment. It was added accordingly: “Other reasons and channels for joining, such as randomly walking past the fitness center, accounted for 16.3% of new members in 2016 and only 4.7% in 2022.” (lines 312-313).
In addition, I suggest you improve the RESULTS section by inserting a more comprehensive comparison between subject types (i.e. men vs women, old vs young, more ageing men vs more aged women, young men vs young women etc.) These comparisons are essential to underline better the differences between male and female users in each age range.
Answer: Thank you for your comment. Please note that the data is already disaggregated by gender and age. We are aware that showing the interactions of multiple independent variables would have added value to our work. However, to perform such an analysis, we would have had to use parametric statistics (i.e., two-way ANOVA), which are not appropriate in this case due to the nature of the data (nominal and ordinal). Therefore, we cannot perform such an analysis and are limited to performing the analysis for each independent variable separately. For this reason, we have also added this to the study limitations (lines 662-665). However, to "strengthen" our results, we have added Table 3. with comparisons between young men vs. young women and adult men vs. adult women in a separate analysis, as noted in your comment (lines 373-379).
DISCUSSION:
Please add in the discussion section a more comprehensive analysis of the comparison among subject types (i.e. men vs women, old vs young, more ageing men vs more aged women, young men vs young women etc.). Eventually, if these further analyses do not provide useful data, please improve the present information about this aspect. Understanding the differences (if any) between the different user types may be very interesting and can help your manuscript increase his interest.
Answer: Added accordingly (lines 577-583).
I hope that my suggestions will be helpful to you.
English is fine and minor editing of the English language is required. However, I am not a mother language reviewer. Thus, I suggest you ask for a mother language consultancy to ensure an excellent English revision.
Answer: We appreciate your constructive comments. We strongly believe that your comments have fundamentally improved our manuscript, making it suitable for publication. Please note that the latest version of the manuscript has been reviewed by a native speaker, so minor corrections have been made throughout the text (marked with the "Track Changes" feature).
Reviewer 2 Report
A well-documented study, even though of regional importance. The article is well-written and easy to read. The data is offered by both males and females which I find very important. Why people engage in fitness centers and why they continue is worthy of study from both a marketing perspective as well as a health fitness perspective. I am guessing this study would be important for the country of Slovenian as well as other countries of the same size. Other than one typo in the abstract - while for white, the article has few errors that I can discern.
Author Response
A well-documented study, even though of regional importance. The article is well-written and easy to read. The data is offered by both males and females which I find very important. Why people engage in fitness centers and why they continue is worthy of study from both a marketing perspective as well as a health fitness perspective. I am guessing this study would be important for the country of Slovenian as well as other countries of the same size. Other than one typo in the abstract - while for white, the article has few errors that I can discern.
Answer: Thank you for your encouraging comments. Please note that all changes have been made to the main document according to the comments of all reviewers using the "Track Changes" feature.
Reviewer 3 Report
The article is an interesting material and constitutes a case study: the authors study on a sample of the Slovenian population how two concepts of the sustainability of fitness centers are particularized: customer recruitment and retention, and motivation for exercises.
Well argued, logically and coherently presented the theoretical arguments, but also the statistical results obtained, the article adds value to the knowledge on the topic. A series of additions, especially on the research design, can increase its value. There are presented below a series of suggestions:
- reformulation of the title so that it refers to the targeted population segment
- identifying keywords that are much more suggestive and appropriate to the topic (for example, fitness center management, young Slovenians, motivation for physical exercises, etc. instead of youth, general population, Europe, Slovenia)
- formulation and introduction of working hypotheses
- in the "Participants" chapter, it would be useful to enter information about the population in order to argue whether the sample is representative. The reason why the independent variable "geographic location (east/west)" has a disproportionate number of subjects (53/235) should be argued. Also, considering the information presented in the paragraph included in lines 200-208, statistical data should be included regarding the two independent variables referred to: marital status and type of training (this information is not included in the present version). If the information presented in lines 200-208 is not maintained, this paragraph can be eliminated
- for a deeper statistical analysis, a series of data could be introduced that reflects the interaction effects of 2 and 3 independent variables on each dependent variable (the article includes information obtained only to verify the statistical differences of a single independent variable on the dependent variables analyzed).
- in the "Discussions" chapter, on lines 292 and 299, the authors refer for comparisons to certain studies that refer only to students (for example, references 40, 41, 44, 45, 47, 48). I suggest the identification of more appropriate studies for the population segment targeted in the study.
- check the wording from line 34 (""being a member of a fitness center member....)
I suggest checking the expression from - line 34 which creates confusion for the reader ("being a member of a fitness center member...")
Author Response
The article is an interesting material and constitutes a case study: the authors study on a sample of the Slovenian population how two concepts of the sustainability of fitness centers are particularized: customer recruitment and retention, and motivation for exercises.
Well argued, logically and coherently presented the theoretical arguments, but also the statistical results obtained, the article adds value to the knowledge on the topic. A series of additions, especially on the research design, can increase its value. There are presented below a series of suggestions:
Answer: Thank you. Please note that all changes have been made to the main document according to your comments using the "Track Changes" feature.
- reformulation of the title so that it refers to the targeted population segment
Answer: Thank you. We have changed the title accordingly. Now it reads: " Purchase Channels and Motivation for Exercise in the Slovenian Population: Customer Behavior as a Guarantee of Fitness Center Sustainability".
- identifying keywords that are much more suggestive and appropriate to the topic (for example, fitness center management, young Slovenians, motivation for physical exercises, etc. instead of youth, general population, Europe, Slovenia)
Answer: Thank you for the suggestion. Has been changed accordingly.
- formulation and introduction of working hypotheses
Answer: Thank you for the comment. The following was added: ” We cannot hypothesize on the first objective of the study due to inconclusive results in the literature and a variety of variables that have been examined. However, regarding motivation for exercise, we hypothesize that social recognition and competition are more important for male and younger participants, while weight management and appearance are more important for female and older participants.” Moreover, please note that we could not hypothesize about the differences between eastern and western Slovenia due to a lack of similar data.
- in the "Participants" chapter, it would be useful to enter information about the population in order to argue whether the sample is representative. The reason why the independent variable "geographic location (east/west)" has a disproportionate number of subjects (53/235) should be argued.
Answer: We are thankful for raising this critical issue. It is true that the sample was unequal. We even mentioned this in the last sentence of the discussion “Nevertheless, we should limit our conclusions regarding geographical criteria due to the relatively disproportionate sample size” (lines 646-647). It was a sample that we had access to. Please note that this was not the primary goal of our study, yet we added the following sentence to the study limitations: “In addition, the different sample size in Eastern and Western Slovenia could potentially bias the results, somewhat limiting their interpretation” (lines 661-662).
Also, considering the information presented in the paragraph included in lines 200-208, statistical data should be included regarding the two independent variables referred to: marital status and type of training (this information is not included in the present version). If the information presented in lines 200-208 is not maintained, this paragraph can be eliminated
Answer: Thank you for this observation. Since this analysis was made as an addition, we overlooked adding it. Now it is added accordingly (line 275).
- for a deeper statistical analysis, a series of data could be introduced that reflects the interaction effects of 2 and 3 independent variables on each dependent variable (the article includes information obtained only to verify the statistical differences of a single independent variable on the dependent variables analyzed).
Answer: We are very grateful for the proposed analyses. We fully agree that the presentation of the mentioned interactions would have added value to our work. However, to perform the proposed analysis, we would have had to use parametric statistics (i.e., two-way ANOVA), which are not appropriate in this case because of the nature of the data (nominal and ordinal). Therefore, we cannot perform such an analysis and are limited to performing the analysis for individual independent variables separately. However, we have also added this to the study limitations (lines 662-665). We thank you for this valuable observation.
- in the "Discussions" chapter, on lines 292 and 299, the authors refer for comparisons to certain studies that refer only to students (for example, references 40, 41, 44, 45, 47, 48). I suggest the identification of more appropriate studies for the population segment targeted in the study.
Answer: We understand the concern raised, but we believe that comparing our results with the mentioned population does not cause bias in the interpretation of our results. Nonetheless, we have found and added an additional study according to your comment (study 45).
- check the wording from line 34 (""being a member of a fitness center member....)
Answer: Thank you. Corrected accordingly. We appreciate your constructive comments. We strongly believe that your comments have fundamentally improved our manuscript, making it suitable for publication. Please note that the manuscript has been reviewed by a native speaker, so minor corrections have been made throughout the text (marked with the "Track Changes" feature).
Round 2
Reviewer 1 Report
Dear Authors,
I noticed that you followed my suggestions and have improved the manuscript according to them. So now, I believe your manuscript fits your study's meaning better.
Great Job!
English quality is fine. Maybe minor revisions or grammar checks are needed. However, I am not a mother language referee. Therefore, I suggest you use a mother language writer to fix possible minor revisions or grammar checks.